# Replication Activities of Major 5′ Terminally Deleted Group-B Coxsackievirus RNA Forms Decrease PCSK2 mRNA Expression Impairing Insulin Maturation in Pancreatic Beta Cells

**DOI:** 10.3390/v14122781

**Published:** 2022-12-13

**Authors:** Domitille Callon, Aurélien Guedra, Anne-Laure Lebreil, Laetitia Heng, Nicole Bouland, Paul Fornès, Fatma Berri, Laurent Andreoletti

**Affiliations:** 1Cardiovir EA-4684, Faculty of Medicine, University of Reims Champagne Ardennes, 51 Rue Cognacq Jay, 51097 Reims, France; 2Pathology Department, Academic Hospital of Reims, Robert Debré, 51100 Reims, France; 3Virology Department, Academic Hospital of Reims, Robert Debré, 51100 Reims, France

**Keywords:** group-B coxsackievirus, pancreatitis, type 1 diabetes, 5′ terminal deletion, insulin

## Abstract

Emergence of 5′ terminally deleted coxsackievirus-B RNA forms (CVB-TD) have been associated with the development of human diseases. These CVB-TD RNA forms have been detected in mouse pancreas during acute or persistent experimental infections. To date, the impact of the replication activities of CVB-TD RNA forms on insulin metabolism remains unexplored. Using an immunocompetent mouse model of CVB3/28 infection, acute and persistent infections of major CVB-TD populations were evidenced in the pancreas. The inoculation of mice with homogenized pancreases containing major CVB-TD populations induced acute and chronic pancreatic infections with pancreatitis. In the mouse pancreas, viral capsid protein 1 (VP1) expression colocalized with a decrease in beta cells insulin content. Moreover, in infected mouse pancreases, we showed a decrease in pro-hormone convertase 2 (PCSK2) mRNA, associated with a decrease in insulin plasmatic concentration. Finally, transfection of synthetic CVB-TD50 RNA forms into cultured rodent pancreatic beta cells demonstrated that viral replication with protein synthesis activities decreased the PCSK2 mRNA expression levels, impairing insulin secretion. In conclusion, our results show that the emergence and maintenance of major CVB-TD RNA replicative forms in pancreatic beta cells can play a direct, key role in the pathophysiological mechanisms leading to the development of type 1 diabetes.

## 1. Introduction

Type 1 diabetes (T1D) is a chronic autoimmune disease characterized by reduced insulin production and caused by a specific immune-mediated loss of insulin-producing beta cells [1]. Etiologically, the initiation and progression of T1D occurs as a consequence of a complex interplay between genetic, immunological and environmental factors [1]. Coxsackie-B viruses (CVBs), which belong to group-B enteroviruses, are the most commonly identified viruses in epidemiological, clinical and histopathological studies of T1D [2,3,4]. CVBs display a tropism for beta cells, and the expression of the Coxsackie and adenovirus receptor (CAR, CXADR), one of the key entry receptors for the CVB strains, has been demonstrated in beta cells [5,6]. Moreover, both enteroviral RNA and the viral protein 1 (VP1), have been detected at low levels in the pancreases of people with T1D at a higher frequency than in control subjects [2,7].

CVBs are ubiquitous human pathogens transmitted through fecal–oral or respiratory routes. CVBs are non-enveloped viruses, with a single-stranded positive-sense RNA genome of approximately 7,400 nucleotides, flanked on the 5′ end by a highly conserved untranslated region (5′ UTR), crucial for the initiation of the viral replication and translation activities. CVB’s 5′ end is known to contain various immunogenic sequences or secondary structures [8]. 

CVBs with 5′ end deletions (CVB-TD) have been detected in the murine pancreas during chronic infection without any detectable cytopathic effect [4,9]. These CVB-TD populations are known to maintain low-grade infection in murine pancreases and in human or mouse hearts [9,10]. We recently showed that CVB-TD maintenance in the pancreases of mice is associated with a decrease in insulin content [11]. These findings suggest that CVB-TD is associated with the development of acute and chronic pancreatitis or T1D in mouse models. Moreover, these reports suggest that these CVB-TD forms could be generated early and selected during natural infection. The emergence of these low-level replicating viruses can explain how the CVB genome and VP1 are detectable in beta cells of individuals with both recent-onset and longer-duration T1D. 

Many mechanisms of viral pathogenesis of T1D have been suggested [12]. One hypothesis is the stimulation of autoimmunity during CVB infection, with the destruction of pancreatic β-cells [13]. Autoantibodies against islets have been detected during CVB infections [14]. This observation could be explained by a molecular mimicry between the amino acid sequences of the CVB viral protease 2C and Glutamic Acid Decarboxylase 65 (GAD65) [15]. Other autoantigens have been reported in patients with T1D, such as against tyrosine phosphatase IA-2 or proinsulin [13,14,16]. Infection of pancreatic β-cells with CVB can result in the diffusion of cellular epitopes, which may stimulate nearby autoreactive cells. Persistent viral replication and a low-grade local inflammatory response could also be involved in the recruitment of autoreactive cytotoxic T cells to the pancreatic islets [17,18]. In T1D patients, only a small number of autoreactive cells have been detected around or in the islets; thus, the cellular inflammatory response alone might be insufficient to cause T1D [16].

An increase in pro-insulinemia in patients with T1D has also been reported, and can play a role in the onset of pancreatic autoimmunity and T1D [19]. In vitro, pro-hormone convertase 2 (PC2) mRNA expression (PCSK2 gene), an enzyme responsible for the processing of proinsulin to insulin, has been shown to be downregulated after persistent CVB4 infection [20]. Thus, a decrease in PCSK2 expression could impair the maturation of proinsulin and lead to low insulinemia, and an increase in pro-insulinemia, followed by pancreatic autoimmunity. A decrease in PCSK2 expression could be the consequence of the viral protease activity, notably the cleavage of transcription factors by protease 3C [21,22].

To date, it remains unknown whether CVB-TD RNA forms are involved in the acute and chronic infection of beta cells or its impact on insulin maturation. In the present study, we investigated the dynamics of natural emergence of CVB-TD RNA populations in the pancreas, as well as pancreatic pathology development during CVB3/28-induced pancreatitis in DBA/2J mice. To delineate the viral mechanisms of insulin impairment, we investigated the direct effect of CVB-TD50 RNA forms, using transfection of synthetic viral RNA, on insulin secretion and maturation in rodent beta cells. Using in vivo and in vitro experiments, we investigated the link between CVB-TD50 emergence, viral replication and translation activities and insulin metabolism impairment in rodent beta pancreatic cells.

## 2. Materials and Methods

### 2.1. Cells, Virus Strain and Reagents

HeLa cells were grown in minimum essential media (MEM), supplemented by 5 mL penicillin–streptomycin (PS) (Gibco^®^, Paris, France), 5 mL L-Glutamine and 500 mL 10% Fetal Bovine serum (FBS) (ThermoFisher^®^, Illkirch-Graffenstaden, France). INS-1-E1 (INS-1) cells were a kind gift from Dr. Bertrand Blondeau and Dr. Ghislaine Guillemain (Sorbonne Université, INSERM UMRS938, Centre de Recherche Saint-Antoine, Paris) and grown in Roswell Park Memorial Institute medium (RPMI1640) supplemented by 1% PS, 10% FBS, 1 μM Pyruvate (1 M) and 50 μM β-mercapto-ethanol. The integrity of INS-1-E1 cells (insulin production ability) was validated. Cells were initially negative for mycoplasma, then their status was regularly verified by a mycoplasma test (PlasmoTest™, InvivoGen, Toulouse, France). The CVB3/28 virus strain was a gift from Dr. Steve Tracy (University of Nebraska, Lincoln, NE, USA). 

### 2.2. Virus Production and Titration

HeLa229 cells were seeded at 0.8 × 10^6^ cells per well of tissue culture platelet (6 wells) and then incubated at 37 °C overnight. The next day, cell confluence was evaluated at 1.5 × 10^6^ cells per well. Based on previous experience, cells were then infected with CVB3/28 at a multiplicity of infection (MOI) of 10^−3^ in free-serum MEM. At 48 h post-infection, the supernatant was collected and then clarified using low-speed centrifugation, and virus particles were then measured by plaque-forming-unit assay (PFU). Briefly, HeLa229 cells were infected with CVB3/28, creating serial dilutions, for 1 h at 37 °C. After viral adsorption, the inoculum was harvested, and cells were overlaid with medium containing 2% agarose and incubated at 37 °C. After 3 days, viral plaques were visualized using bromophenol blue staining. The viral titers were expressed as plaque-forming unit per 1 mL (PFU/mL). The deleted 5′ UTR viral sequences were obtained using a two-step PCR approach to delete the 5′ nucleotides and add a hammerhead ribozyme upstream of the viral sequence. The 5′ TD viral RNA were synthesized using a MEGAscript™ T7 Transcription Kit (Invitrogen^®^, Waltham, MA, USA) and purified using a MEGAclear™ Transcription Clean-Up Kit (Invitrogen^®^).

### 2.3. Ethic Statement

Experiments were performed according to recommendations of the “National Commission of Animal Experiment (CNEA)” and the “National Committee on the Ethical Reflection of Animal Experiments (CNREEA)”. The protocol was approved by the committee of animal experiments of the Reims Champagne-Ardenne (accreditation number 56) and then by Ministry of Advanced Education and research (permits number #5463-201704171500306). All efforts were made to minimize suffering in accordance with the Guide for the Care and Use of Laboratory Animals of the Direction des Services Vétérinaires, the French regulations to which our animal care and protocol adhered. Notably, mice were weighed every day and we used a mouse grimace scale to assess suffering.

### 2.4. Infection of Mice

Male DBA/2J mice, purchased from Charles River laboratory^®^ and housed under pathogen-free conditions, were acclimated for one week before the experimentations (only handling). Two to five mice were housed in one cage, with free access to food and water. Mice were randomly assigned to either infected or non-infected groups. Then, mice (5 weeks old) were inoculated intraperitoneally with 10^6^ PFU/mice (CVB3/28) or clarified homogenized-pancreas; uninfected mice were inoculated with 150 µL of saline. To determine survival rates upon infection, loss of body weight was monitored daily for 28 days post-infection, and mice were euthanized if they had ≥20% loss of their initial body weight, according to our protocol. Then mice were anesthetized with ketamine/xylazin (42.5/5 mg/kg) and euthanized by cervical dislocation before organ collection at various pre-fixed times points post-infection (8, 16, 24, 48 and 72 h post infection; 7, 14 and 28 days post-infection). Infected or uninfected mice were sacrificed at the indicated times, complete autopsies were performed, and pancreases were taken and either fixed in formaldehyde 4% for histology or homogenized in 600 µL of PBS solution with TissueLyser LT (QIAGEN^®^, Courtaboeuf, France).

### 2.5. Histology, Immunohistochemistry and Immunofluorescence

For each sample, four-µm-thick paraffin sections were cut at 3 levels. The slides were stained with hematoxylin, eosin and safran (HES, Pathology department, Academic Hospital of Reims, France). The slides were examined by a pathologist for cellular infiltration and necrosis, blinded to the group assigned (0: no inflammation, no necrosis; 1: sparse inflammatory cells, no necrosis; 2: inflammatory cells infiltrates with necrosis; 3: complete inflammatory necrosis of the pancreas). In addition, immunostaining (immunohistochemistry) was performed for CD3 (1/1000; Dako, Santa Clara, CA, USA, A0452), CD68 (1/100; Dako, M0814) and VP1 (1/1000; Dako, 5D8/1) as previously described [23,24]. For VP1 immunohistochemistry, a NovoLink Polymer Detection Systems kit (Leica Biosystems, Newcastle, UK) was used, with endogenous peroxidase activity blocking and mouse antigen blocking. For immunofluorescence assays, after using a blocking reagent for mouse antigenicity, 1:200-diluted rabbit anti-insulin (ABclonal, Woburn, MA, USA, A2090), and 1:500-diluted mouse anti-VP1 (Dako, 5D8/1) were used for the staining of cells and fixed pancreas sections. Secondary antibodies were conjugated with AlexaFluor514 (anti-rabbit, 1/200) and AlexaFluor633 (anti-mouse, 1/400) (Thermofischer, Waltham, MA, USA). DAPI (1/10,000; Sigma, St. Louis, MO, USA) was used to stain cell nuclei. For further analyses, positive islets for VP1 and insulin were selected using QuPath software. Confocal images were acquired by using a Zeiss confocal microscope. Intensities and colocalization (Pearson correlation) were quantified by using QuPath 0.2.3 software (University of Edinburgh, Edinburgh, UK) and JACoP plugin of ImageJ.

### 2.6. Protein and Viral Titer Quantification

Pancreases of euthanized mice were homogenized in 600 µL of PBS solution with Tissue-Lyser LT (QIAGEN^®^, Hilden, Germany), and then centrifuged at 12,000 rpm for 15 min at 4 °C. Blood was centrifuged at 10,000 rpm for 10 min at 4 °C to obtain plasma. The supernatant or the plasma were used for: (1) measurement of proteins by enzyme-linked immunosorbent assay, such as insulin, proinsulin (Elabscience^®^, Houston, TX, USA) and GAD/IA2 autoantibodies (MyBiosource^®^, San Diego, CA, USA); (2) determination of virus particles. Briefly, HeLa229 cells were infected with homogenized pancreas, with serial dilutions, for 1 h at 37 °C. After viral adsorption, the inocula were harvested, and cells were overlaid with medium containing 2% agarose, then incubated at 37 °C. After 3 days, viral plaques were visualized using bromophenol blue staining. Viral titers were expressed as plaque-forming unit per 1 mg of total protein (PFU/mg). The measurement of total protein concentration in each supernatant was performed by the Bicinchoninate Acid Method (Sigma Aldrich^®^, Saint-Quentin-Fallavier, France) according to the manufacturer’s instructions.

### 2.7. In Vitro Infection and Transfection

INS-1 were seeded at 2 × 10^5^ cells per well of tissue culture platelet (24 wells) and then incubated at 37 °C overnight. The next day, cells were infected for 1 h at 37 °C with CVB3/28 at the indicated MOI (1 or 10), in free-serum RPMI1640. After viral adsorption, the inocula were harvested and cells were washed with RPMI1640 three times, before an incubation with free-serum RPMI1640 at 37 °C. Cells and supernatants were collected at the indicated times. 2.0 × 10^5^ INS-1 cells per well were seeded in 24-well plates (Nunclon delta surface, Thermo Scientific). The plates were incubated overnight at 37 °C. The transfection mixtures (0.1 mL) consisted of 1 µg of synthetic RNA (full-length or TD50-CVB3 RNA) and 1 μL of Lipofectamine 2000 (Thermo Scientific) in Opti-MEM Glutamax. The mixtures were incubated for 20 min at room temperature. Cells were washed with DPBS (Thermo Scientific). Then, 100 μL of the mixture was added to the cells, and plates were incubated at 37 °C for 2 h. Cells were then washed three times, and free-serum RPMI1640 was added for incubation at 34 °C. Media were collected at the indicated times for viral load measurements, and cells were used for a glucose-stimulated insulin secretion assay after 24 h of incubation.

### 2.8. Glucose-Stimulated Insulin Secretion

For insulin secretion measurements, INS-1 cells were starved for 1 h in Krebs Ringer Buffer (KRB, Alfa Aesar^®^, Haverhill, MA, USA) containing 2 mM glucose (Alfa Aesar). After starvation, medium was exchanged for KRB containing 25 mM glucose in order to measure glucose-stimulated insulin secretion for a 1-h incubation period. The amount of secreted insulin was measured in the media (ELISA, Elabscience^®^, Houston, TX, USA), and INS-1 cells were subsequently dissolved in RIPA buffer to measure total protein content. Secreted insulin was normalized to total protein content (ng/mg of total protein).

### 2.9. Western Blot

SDS-polyacrylamide gel electrophoresis was performed on 4–15% Mini-PROTEAN TGX Stain-Free gels (BioRad; #4568085) or NuPAGE 3–8% Tris-Acetate gels (Invitrogen, #EA03752BOX), then transferred to cellulose membranes (GE Healthcare, Little Chalfont, UK) with Trans-Blot^®^ SD Semi-Dry Transfer Cell. The following antibodies were used: anti-VP1 (1/500, Dako, 5D8/1), anti-eIF4G (1/1000, #2469S, Cell Signaling Technology, Danvers, MA, USA) and monoclonal anti-β-actin antibody (1/1000, #3700, Cell Signaling Technology). HRP-coupled anti-mouse (1/5000, NA9310V, GE Healthcare) and anti-rabbit (1/10,000, #7074S, Cell Signaling Technology) were used as secondary antibody. Peroxidase activity was visualized with an ECL Plus Western Blotting Detection System (#RPN2132, GE Healthcare).

### 2.10. Total RNA Isolation and qRT-PCR

The homogenized pancreases were subjected to digestion with proteinase K (Merck^®^, MO, USA). The total RNA was isolated with the TRIzol RNA Isolation Reagents^®^ (Thermo Fisher^®^) method according to the manufacturer’s instructions. RNA were stored at −80 °C. The viral RNA copy number was quantified by RT–quantitative PCR with a StepOne plus realtime PCR system (ThermoFisher Scientific^®^) as described previously [11], using primers amplifying the CVB3/28: sense (5′-CAC ACT CCG ATC AAC AGT CA-3′), antisense (5′-GAA CGC TTT CTC CTT CAA CC-3′) and probe FAM/TAMRA (5′-CGT GGC ACA CCA GCC ATG TTT-3′) (Eurogentec^®^, Seraing, Belgium).

### 2.11. Reverse Transcription-Quantitative PCR Using SybrGreen^®^ (RT-qPCR)

RNA was extracted from pancreas lysates with TRIzol Reagent^®^ (Invitrogen, Life Tech-nologies) and reverse-transcribed using SuperScript™ II Reverse Transcriptase (RT) (Invitrogen, Life Technologies, Saint-Aubin, France), following the manufacturer’s instructions. cDNA was subjected to PCR using PowerUp™ SYBR™ Green Master Mix (2X) (ThermoFisher Scientific^®^). The primers used for PCR detection were as follows: for mouse insulin, forward primer: 5′-CGTGGCTTCTTCTACACACCCA-3′, reverse primer: 5′-TGCAGCACTGATCCACAATGCC-3′; for mouse PCSK2, forward primer: 5′-ACCTCTTTGGCTACGGAGTCCT-3′, reverse primer: 5′-TTGAGGGTCAGTACCAGCTTCC-3′; and for mouse ISG15, forward primer: 5′-CATCCTGGTGAGGAACGAAAGG-3′, reverse primer: 5′- CTCAGCCAGAACTGGTCTTCGT-3′. Primers specific to mouse GAPDH (forward primer: 5′-CATCACTGCCACCCAGAAGACTG-3′, reverse primer 5′-ATGCCAGTGAGCTTCCCGTTCAG-3′) were used as the internal control. PCR was carried out in StepOnePlus Real-Time PCR Systems (ThermoFisher Scientific^®^), which were programmed as follows: denaturation step at 95 °C for 15 s, annealing/extension step at 65 °C for 1 min (40 cycles) before melting curve. Results were analyzed with the ΔΔCT method, where CT is threshold cycle, and normalized to GAPDH mRNA. Data are represented as levels of mRNA relative to those of the mock transfected control samples, and are displayed as the means ± SD of results from at least three independent in vitro experiments.

### 2.12. Rapid Amplification of cDNA Ends-PCR (RACE-PCR)

Viral RNA (200 ng) was reverse transcribed using a Superscript II kit (Invitrogen, ThermoFisher^®^) with 400 nM AvCRev. The 5′ extremity of cDNA was then ligated with a T4 DNA Ligase (Ambion, ThermoFisher^®^) and incubated for 24 h at 16 °C. Positive cDNA was amplified by a classical PCR reaction and amplicons were then sized and quantified by Bioanalyzer High Sensitivity DNA Analyses (Agilent^®^, Santa Clara, CA, USA) as previously described [11,25]. Due to the sensitivity of the analyzer, results with deletions under 8 nucleotides were waived, and the sequences were classified as full-length.

### 2.13. Positive and Negative-Strand RNA Ratio

Negative-strand RNA was isolated from total RNA by annealing a biotinylated negative-strand-specific primer and binding to streptavidin-labeled magnetic beads (Invitrogen^®^, Life Technologies, Saint-Aubin, France), and was then quantified by RT-qPCR as previously described [26]. The positive- to negative-strand viral RNA ratio was then determined using the following calculation: (total EV RNA—negative strand EV RNA)/negative strand EV RNA.

### 2.14. Single and Double-Strand RNA Ratio

Double-strand EV-B RNA was isolated by digesting single-strand RNA with recombinant bacterial RNase A (ThermoScientific^®^). A mix of 200 ng of total RNA, 1 μL of RNase and buffer solution of NaCl at 0.3 M was incubated for 10 min at 25 °C. The double-strand viral RNA copy number was quantified with a one-step real-time RT-PCR assay using serial dilutions of the transcripts for the generation of the standard curves. The single-strand to double-strand viral RNA ratio was then determined using the following calculation: (total EV RNA—double-strand EV RNA)/double-strand EV RNA.

### 2.15. Replication Capacities, Progeny and Encapsidation Assessment

HeLa cells were seeded at 1.5 × 10^5^ cells per well of tissue culture platelet (24 wells) and then incubated at 37 °C overnight. The next day, cell confluence was evaluated, based on previous experience, and cells were infected with 300 μL of homogenized pancreas supernatants for 1 h at 37 °C. Encapsidation of these viral RNA forms was assessed by challenging viral entry capacity with or without proteinase K treatment (Merck^®^), which digests all proteins, including viral proteins in the sample. After viral adsorption, the inocula were collected, and cells were overlaid with free-serum MEM and incubated at 37 °C. For entry capacity assessment, cells were harvested after the viral adsorption. The viral RNA copy number was quantified by RT–quantitative PCR with a StepOne plus realtime PCR system (ThermoFisher Scientific^®^) as previously described [11], using primers amplifying the CVB3/28: sense (5′-CAC ACT CCG ATC AAC AGT CA-3′), antisense (5′-GAA CGC TTT CTC CTT CAA CC-3′) and probe FAM/TAMRA (5′-CGT GGC ACA CCA GCC ATG TTT-3′) (Eurogentec^®^, Seraing, Belgium). With proteinase K treatment, no viral RNA was detected in HeLa cells. We calculated the percentages of viral entry as follow: % = (viral load in HeLa cell after 1 h incubation/viral load in the inoculum).

For replication activity, the supernatant and cells were harvested separately after 72 h. The viral RNA copy number was quantified by RT–quantitative PCR with a StepOne plus realtime PCR system (ThermoFisher Scientific^®^) as previously described [11], using the same primers. For viral titer measurement, HeLa 229 cells were infected with CVB3/28, creating serial dilutions, for 1 h at 37 °C. After viral adsorption, the inocula were harvested, and cells were overlaid with a medium containing 2% agarose, then incubated at 37 °C. After 3 days, viral plaques were visualized using bromophenol blue staining. The viral titers were expressed as plaque-forming unit per 1 mL (PFU/mL).

### 2.16. Statistical Analysis

Mann–Whitney’s test was used for viral organ data, in vitro viral data and ELISA results. Spearman’s correlation test was performed. The observed differences were considered significant at a *p* value ˂ 0.05. The number (*n*) of animals used in the experiment is specified in the figure legend. All statistical analyses were performed using GraphPad Prism 7 (Prism^®^, San Diego, CA, USA).

## 3. Results

### 3.1. Early Emergence and Maintenance of 5′ Terminally Deleted RNA Forms during CVB3/28-Induced Pancreatitis Are Associated with a Decrease in Viral Replication Levels

To investigate early dynamics of the 5′ terminally deleted (CVB-TD) viral population’s emergence in the mouse pancreas, we first quantified the total CVB load in the pancreas from 8 h to 28 days post infection (H or DPI) (Figure 1A). Pancreatic viral loads exhibited an increase until 48 HPI (Figure 1A). Total pancreatic viral loads exhibited a decrease until 28 DPI, but remained detectable with values under or near a threshold of 10^3^ genome copies (gc) per microgram of extracted RNA, as previously described (Figure 1A) [11]. Viral progeny was measured by classical PFU assay, demonstrating a peak at 72 HPI, with an absence of detectable infectious particles at 28 DPI (Figure 1B). To determine the dynamics of emergence of CVB-TD populations, we used a previously validated RACE-PCR to quantify CVB-TD and full-length (CVB-FL) viral populations. Two subpopulations of CVB-TD forms were characterized: one deleted from 8 to 36 nucleotides (nt) and one deleted from 37 to 50 nt (Figure 1C,D). As depicted in Figure 1C,D, CVB-TD forms emerged as early as 8 HPI (Figure 1C,D). These CVB-TD populations were initially associated with CVB-FL forms, which became undetectable after 28 DPI (Figure 1D). We showed that there were significantly more CVB-TD than CVB-FL forms at 3 DPI. The viral load peak during acute pancreatitis (*p* < 0.0001) and that the CVB-TD/FL ratio increased significantly between 3 and 7 DPI (*p* < 0.0001) (Figure 1C–E). Our results evidenced early emergence of CVB-TD populations during acute pancreatitis, without any detectable CVB-FL RNA forms after 28 DPI.

To explore the potential association between viral genomic replication and CVB-TD populations in vivo, we used correlation analyses. Interestingly, we showed a negative correlation between the proportions of CVB-TD forms and total viral loads (R^2^ = 0.79, *p* < 0.0001) between 2 and 7 DPI (Figure 1F). These results showed that, in vivo, CVB-TD emergence results in a decrease in replication activities in pancreatic cells.

### 3.2. Viral Characteristics of CVB-TD RNA Forms Detected in Pancreases of DBA/2J Mouse

To further evaluate the RNA genomic replication capacities of CVB populations in the mouse pancreas, we determined the (+)/(−) viral RNA and the single/double-strand (ss/ds) viral RNA ratio (Table 1). High (+)/(−) and ss/ds RNA ratios were evidenced at 3 DPI, indicating high levels of viral genomic replication (Table 1). By contrast, after 14 DPI, a decrease in (+)/(−) and ss/ds RNA ratio indicated impaired viral genomic replication (Table 1). Moreover, viral progeny was high at 3 DPI, low at 14 DPI and undetectable at 28 DPI (Figure 2B). We showed that CVB populations at 3 DPI were encapsidated (73%) (+) ssRNA forms, whereas at 14 and 28 DPI, CVB populations were non-encapsidated (9.2% and 2%, respectively) (−) or dsRNA forms (Table 1). Overall, we evidenced high levels of viral genomic replication with production of native infectious viruses in the pancreas before 7 DPI (Figure 1A,B and Table 1). After 28 DPI, we showed that low viral genomic replicative and protein synthesis activity levels, which are hallmarks of viral persistence, were associated with an absence of a detectable viral progeny in the pancreas (Figure 1A,B, and Table 1).

### 3.3. CVB-TD Populations from Homogenized Infected Pancreases Induce an Acute and Persistent Infections in Islet Beta Cells with Viral Capsid Protein 1 Expression and Inflammatory Cell Infiltrates

To assess the ability of CVB-TD populations to infect beta cells in vivo, we inoculated DBA/2J mice with clarified, homogenized pancreas, sampled at 3 (1.35% of CVB-FL; 6.76% of CVB-TD 8–36 nt and 91.89% of CVB-TD 37–50 nt), 14 (15.64% of CVB-FL; 11.18% of CVB-TD 8–36 nt and 73.18% of CVB-TD 37–50 nt) or 28 DPI (28.46% of CVB-TD 8–36 nt and 71.54% of CVB-TD 37–50 nt) from CVB3-infected mice (Figure 2A). Following an infection running from 1 to 28 days, we showed that pancreatic CVB-TD populations of infected mice at 3, 14 and 28 DPI had reached the pancreas and demonstrated stable replication activities, without detectable infectious particles at 28 DPI (Figure 2B,C). As expected, CVB populations from 3-DPI homogenized pancreas demonstrated higher replication activities than those from 14 and 28 DPI samples at the acute phase, with no significant variations thereafter (*p* < 0.0045; Figure 2B). Despite the absence of infectious particles in 28-DPI homogenized pancreas supernatants, their intra-peritoneal inoculation induced sparse inflammatory foci or necrosis surrounding Langerhans islets, with VP1 expression in islet cells (Figure 2D). CVB-TD RNA populations remained detectable until 28 DPI in all pancreases, even after 28-DPI homogenized pancreas inoculation, without CVB-FL RNA forms (Figure 2E).

Together, these results show that pancreatic CVB-TD RNA populations from previously CVB3/28-infected pancreases can infect and replicate in pancreases in vivo, with viral protein expression until 28 days post infection, despite the absence of CVB-FL forms or detectable infectious particles.

### 3.4. Viral Replication and Translation Activities of CVB3/28 RNA Populations Impair the Insulin Content and Maturation in Islet Cells of Immunocompetent Mice

To investigate the impact of CVB3/28 infection on endocrine pancreases, we performed confocal analyses allowing us to explore beta cells’ insulin expression. The immunostaining of mouse pancreas sections showed a co-expression of insulin and VP1 proteins in islet cells, increasing between 3 and 28 DPI, as shown on fluorograms (Figure 3A). The insulin/VP1 intensity ratio was elevated until 7 DPI, and decreased at 28 DPI (*p* < 0.021; Figure 3B). The Pearson correlation coefficient increased at 7 and 28 DPI in beta cells (*p* = 0.008; Figure 3C).

Because we observed a decrease in insulin content in CVB3/28-infected pancreases, we examined the insulin secretion and maturation in mice infected with CVB3/28 or with CVB-TD RNA populations obtained from previously infected pancreases. We first semi-quantified the expression of Ins1 gene mRNA, coding for the preproinsulin (Figure 4A). We showed that there was no decrease nor increase in Ins1 gene mRNA expression following CVB3/28 infection in mice at 3 or 28 DPI (Figure 4A). Proinsulin concentration in the plasma increased during acute and chronic pancreatitis (*p* = 0.0079; Figure 4B). An insulinemia peak was observed at 3 DPI, with a significant decrease thereafter (*p* = 0.0079; Figure 4C). Proprotein Convertase Subtilisin/Kexin Type 2 (PCSK2, gene coding for the Pro-hormone Convertase 2, the enzyme processing proinsulin into insulin) mRNA expression levels decreased between acute and chronic infection with CVB3/28 as well as with CVB-TD RNA populations (*p* < 0.0022; Figure 4D). Autoantibodies against GAD65 and IA2 were elevated during infection with CVB3/28 or inoculation of CVB-TD RNA populations, at both the acute and chronic stages (*p* < 0.033, Figure 2E).

Together, these results show that CVB3/28 infection and the inoculation of CVB-TD RNA populations in mice decrease the insulin secretion, possibly by an impairment of proinsulin to insulin maturation, and induce the production of autoantibodies against islet cells during chronic infection.

### 3.5. CVB-TD50 RNA Transfection into INS-1-E1 Cells Decreases PCSK2 mRNA Expression and Impairs Insulin Secretion

Because we showed an impairment of insulin maturation in mice infected with CVB3/28 and CVB-TD RNA forms (Figure 3 and Figure 4), we performed infection and transfection of the beta cell line of rodent INS-1, with the CVB3/28 strain at a MOI of 1 and with CVB-FL or CVB-TD synthetic RNA, respectively. First, we showed that CVB3/28 or synthetic RNA of CVB-FL and CVB-TD50 had active replication activity in INS-1 cells at 24 h post-infection (*p* < 0.02; Figure 5A–C). To investigate the translation efficiency of CVB3/28 infection or CVB-FL or CVB-TD50 transfection in INS-1 cells, we performed Western blot analysis. We showed that viral capsid protein (VP1) was expressed at 24 h post infection/transfection with CVB3/28 or CVB-FL/TD50 in INS-1 cells, thus demonstrating efficient translation activities (Figure 5B–D). Moreover, we showed that CVB3/28 infection of INS-1 cells results in the cleavage of host factor eIF4G by viral proteinase, as previously shown (Figure 5B) [10]. We used Poly (I:C) transfection of INS-1 cells to induce an inflammatory response, mimicking a viral infection without viral protease activity [27]. PCSK2 mRNA expression decreased at 24 h post-infection or after CVB-FL or CVB-TD transfection, but not after Poly (I:C) transfection (*p* = 0.002; Figure 5E). As shown in Figure 5E, ISG15 mRNA expression increased at 24 h post transfection with Poly (I:C), as previously shown [27], but not after CVB3/28 infection or CVB-FL/TD50 transfection (*p* = 0.002, Figure 5F). INS-1 cells infected with CVB3/28 responded like uninfected controls did, with respect to the ability to release insulin below 2 mM glucose conditions (Figure 5G). However, the insulin release in response to high glucose (25 mM) stimulation was hampered in CVB-infected INS-1 cells compared to mock-infected cells (*p* < 0.0001; Figure 5G). INS-1-cells transfected with CVB-FL or TD50 RNA forms did not increase their insulin secretion in response to high glucose stimulation, similarly to CVB3/28-infected cells (Figure 5F). Interestingly, the insulin response to high glucose stimulation was maintained following Poly (I:C) transfection, despite the inflammatory response mimicking a viral infection (*p* = 0.042, Figure 5G). Taken together, these results showed that the replication and translation activities of CVB-TD50 RNA forms decrease PCSK2 mRNA expression and the insulin response of rodent beta cells, possibly by proteinase-induced cleavage activities.

## 4. Discussion

Emergence of 5′ terminally deleted coxsackievirus-B RNA forms (CVB-TD) have been associated with the development of acute or chronic pathologies in humans and in experimental mouse models [2,4]. CVB-TD has been previously detected in the heart tissues and peripheral blood of patients with acute myocarditis or chronic dilated cardiomyopathy [10,25,28]. In cardiac cells, the CVB-TD RNA population’s infection induced the disruption of dystrophin by sufficiently maintained viral proteinase-induced cleavage activities [10]. Moreover, CVB-TD RNA forms have been detected in mouse pancreases during chronic infection, and have been associated with low insulin content in the pancreas or a decrease in insulin expression [9,11]. Using an immunocompetent mouse model of systemic CVB3/28 infection, we recently showed that a persistent pancreatic infection by these CVB-TD RNA forms was associated with an insulin content decrease [11]. In the present investigation, we designed our successive in vivo and in vitro experiments to explore the direct impact of the replication and translation activities of major CVB-TD RNA forms onto the insulin metabolism in rodent pancreatic beta cells.

We first evidenced the early emergence and maintenance of major CVB-TD populations in pancreases of CVB3/28-infected DBA/2J mice, characterized by low levels of viral replicative activities (Figure 1, Table 1). These results are concordant with the description of such persistent CVB3-TD RNA forms in the hearts of patients with dilated cardiomyopathies or in mouse hearts [10,11,28]. Interestingly, we showed that the inoculation of CVB-TD populations from previously infected pancreases into immunocompetent enterovirus-free mice re-induced successively acute and persistent pancreatic infections, with low viral translation activities, as demonstrated by low VP1 expression levels (Figure 2). These original findings demonstrated the pancreatic tropism of CVB-TD RNA populations and their ability to induce pancreatitis in newly infected mice, showing the pancreatic pathogenicity of these viral forms, as in the classical Koch’s postulates [29].

In mouse pancreases, we showed that CVB-TD RNA populations displayed low replication activity levels, still maintaining persistent infection with translation activity (Figure 2, Table 1). We observed that the VP1 expression colocalized with a decrease in insulin content, as shown by the decrease in insulin staining intensity and the colocalization between VP1 and insulin staining (Figure 3). Such a low-grade infection of beta cells is concordant with VP1 detection in a small number of beta cells of patients with both recent-onset and longer-duration T1D [7,30,31]. Moreover, other pancreatic-type cells were positive for VP1 in this model, either by immunohistochemistry or by immunofluorescence (Figure 2 and Figure 3). These cells could be ductal cells, which have been shown to be infected by various CVBs in vitro [32,33]. In this context, the host and virus interacted without the development of large-scale beta cell lysis. In cases of T1D associated with CVB infection, the emergence of CVB-TD associated with a moderate inflammatory response could result in a chronic persistent infection with sufficient levels of viral protein synthesis activities, reducing the insulin secretion. Interestingly, in infected mouse pancreases, we showed a decrease in pro-hormone convertase 2 (PCSK2 gene, coding PC2 protein) mRNA, associated with a decrease in insulin plasmatic concentration and an increase in proinsulin plasmatic concentration during persistent infection (Figure 4). Taken together, these results obtained in persistently CVB-TD infected murine pancreas indicated an impairment of insulin maturation, which could be the consequence of a decrease in PCSK2 mRNA transcriptional levels (Figure 4). We hypothesized that the impaired processing of proinsulin into insulin could be the direct consequence of viral proteinase cleavage activities of specific transcription/translation host cell factors (Appendix A). Low viral translation activities during the persistent infection of pancreatic beta cells could be sufficient to disrupt host cell proteins, such as eIF4G (eukaryote translation factor 4G) or TBP (TATA box binding protein, a transcription factor), both known to be cleaved by enteroviral proteinases (Figure 1 and Figure 2, Table 1) [10,21,34]. Moreover, DNA hypermethylation was also demonstrated by some authors during persistent CVB4 infection of beta cells in vitro, decreasing host cell transcription levels [20]. Transcriptomic and proteomic approaches are needed to determine the mandatory host cell factors implicated in PCSK2 mRNA production and translation in pancreatic beta cells, which might potentially be cleaved by CVB proteinases.

To assess the link between the replication activities and insulin maturation impairment of major CVB-TD forms, synthetic CVB-TD50 RNA forms were transfected into cultured rodent beta cells (INS-1 cells). Remarkably, we showed that CVB-TD50 RNA forms with replicative and translation activities induced a decrease in PCSK2 mRNA expression, impairing the insulin secretion in response to high glucose stimulation (Figure 5; Appendix A). Recently, it was shown that mRNA expression of insulin and C-peptide concentrations, a cleavage product of insulin processing, were decreased during persistent CVB4-E2 infection of human primary pancreatic ductal cells [33]. Accordingly, other authors reported that PCSK2 mRNA expression was decreased during persistent infection of beta cells with CVB4 [20]. Herein, we showed that CVB3/28 infection and CVB-TD50 RNA forms’ transfection diminished the insulin secretion in response to high glucose concentration in INS-1-E1 cells (Figure 5). Interestingly, the transfection of Poly (I:C) in cultured beta cells increased ISG15 mRNA expression levels without a complete shut-down of the insulin response to high glucose stimulation levels (Figure 5), demonstrating that the antiviral inflammatory response alone was not sufficient to abolish the insulin secretion or its processing. By comparison, transfection of CVB-TD50 RNA forms did not induce an ISG15 mRNA expression increase, but maintained viral translation activities, as demonstrated by Western blotting analyses of VP1 levels (Figure 5). These results indicated that viral replication and translation activities, without an antiviral inflammatory response, could directly impair the insulin secretion or maturation of cultured beta cells. Moreover, we showed that CVB3/28 infection in INS-1-E1 cells resulted in eIF4G cleavage, demonstrating viral proteinase activities (Figure 5). Taken together, these findings suggested that the insulin maturation impairment by CVB-TD50 RNA forms might be a consequence of the viral proteinase activities rather than a consequence of beta cells’ inflammatory responses in CVB-infected rodent beta cells (Figure 5) [10,35].

Molecular mimicry mechanisms between CVB protein epitopes and host beta cell proteins recognized by autoreactive T cells have been reported in several studies [17,36,37]. Possible cross-reactivity with GAD65 or IA-2 epitopes has also been reported [15,37]. The combination of chronic infection of beta cells with low inflammation could enhance autoreactive T cell populations [1,20]. In our model, autoantibodies against GAD65 and IA-2 were increased during acute and chronic infection, even after the inoculation of CVB-TD populations (Figure 4). The low replicating CVB-TD populations could trigger the secretion of such autoantibodies by a viral protein synthesis and chronic viral antigen presentation [15,36]. Moreover, we showed that proinsulin plasma levels, an autoantigen detected during T1D onset, were also increased after acute and chronic CVB3/28 infections in mice (Figure 4) [19]. Further investigations are needed to explore the secretion of CVB-TD-induced autoantibodies against beta cells.

Finally, our results showed that CVB-TD RNA forms could be key players in T1D development. Using our experimental approach applied to a human model, such as the human beta-cell line EndoC-BH1 or hiPSC-derived into beta cells, could increase the knowledge of CVB-TD-induced pathogenesis in T1D. Finally, CVB RNA populations’ clearance could be a potential therapeutic target in T1D patients. Since persistent CVB RNA forms have low replication activities and low encapsidation rates (Table 1), antiviral therapies targeting viral polymerase or encapsidation might be less efficient than potential immunotherapy strategies to restore antiviral innate responses, specifically type I interferon activation pathways [11,38]. Such immunotherapies could achieve viral clearance of CVB-TD RNA forms in pancreatic beta cells, thus limiting the enterovirus-induced development of T1D in the young [39]. Moreover, a primary vaccination against CVB during childhood could also prevent early, acute and persistent infections of CVB-TD RNA forms and subsequent chronic lesions of the endocrine pancreas tissues [11]. Secondary vaccination, after CVB infection, might also prevent re-infection and reactivation of CVB viral RNA replication activities. New emerging mRNA vaccination strategies could target multiple CVB types and induce an innate immune response sufficient to clear acute and chronic enterovirus infections in humans. A CVB vaccine, which is currently undergoing clinical trials in Finland (Provention Bio, Red Bank, NJ, United States), was developed with a focus on the involvement of CVBs in diabetes (PROVENT clinical trial).

In conclusion, our results show that the emergence and maintenance of major CVB-TD RNA replicative forms in beta pancreatic cells can play a direct, key role in the pathophysiological mechanisms leading to the development of T1D. These original findings should stimulate the development of new anti-CVB vaccination strategies to prevent acute infection in the young, or to induce a viral clearance in persistently infected human pancreases.

## Figures and Tables

**Figure 1 viruses-14-02781-f001:**
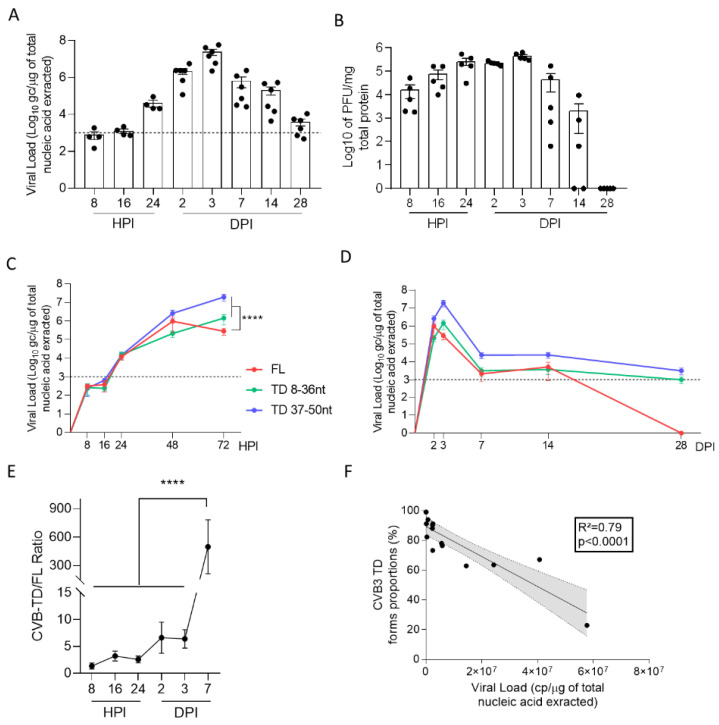
Dynamics of emergence and replication activities of 5′ terminally deleted CVB RNA forms in the pancreas. (**A**) Viral loads were quantified in the pancreas of CVB3/28-infected DBA/2J mice by classic RT-qPCR assay (*n* = 4 to 6, each dot represents one mouse). The dotted line represents the significance threshold of 10^3^ genome copy (gc)/microgram of total nucleic acid extracted. (**B**) Viral titers were measured in the pancreases of CVB3/28-infected DBA/2J mice by classic plaque-forming unit assay (PFU) (*n* = 5, each dot represents one mouse). (**C**,**D**) The respective viral loads of EV-B full-length (FL) and 5′TD were measured using a RACE-PCR method associated with micro-electrophoresis (Agilent^®^) in the pancreases of CVB3/28-infected mice (*n* = 4 to 8). (**A**–**D**) Data represent mean +/− standard error of the mean (SEM). (**E**) 5′ TD/FL ratio in the pancreases of CVB3/28 infected mice from 8 HPI to 7 DPI. Multiple comparison test of the 5′ TD/FL ratio. ****: *p* < 0.0001, multiple comparisons test. (**F**) Linear regression curves and Spearman R coefficients of correlation between viral loads and 5′ TD EV-B proportions (2 to 7 DPI; *n* = 15, each dot represents one mouse). DPI: days post infection. HPI: hours post infection. CVB: coxsackievirus. TD: 5′ terminally deleted. FL: full-length. nt: nucleotides.

**Figure 2 viruses-14-02781-f002:**
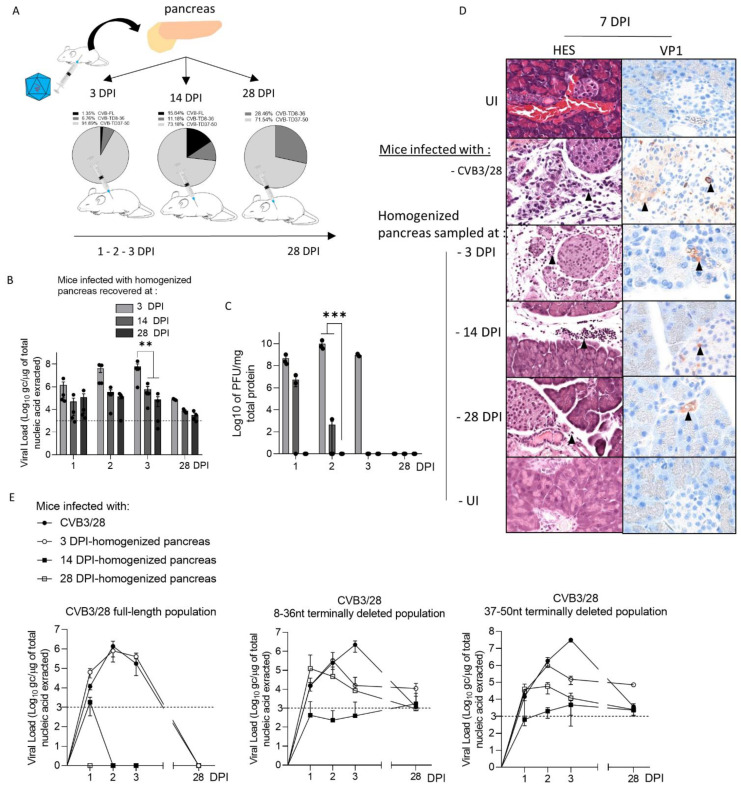
Inoculation of CVB RNA pancreatic populations in DBA/2J mice induced acute and persistent infections with capsid viral protein 1 expression. (**A**) Schematic representation of the experimental procedure. Pancreases were sampled after 3, 14 or 28 days of CVB3/28 infection, and were then homogenized and clarified. Supernatants were inoculated by the intra-peritoneal route to uninfected mice. (**B**) CVB3/28 RNA loads, extracted from pancreases of mice inoculated with 3, 14 or 28 DPI-infected pancreases, were quantified by RT-qPCR (*n* = 4, each dot represents one mouse). (**C**) Infectious CVB3/28 pancreas virus titers at the indicated time points post-infection of mice inoculated with 3, 14 or 28 DPI-infected pancreases (*n* = 4, each dot represents one mouse). (**D**) Histological analysis of pancreases from infected or uninfected mice (7 days post-infection). Thin sections of pancreas obtained from uninfected mice or mice inoculated with homogenized pancreas, sampled at 3, 14 or 28 DPI by CVB3/28. Uninfected (UI) pancreases (original magnification: ×200) were stained with hematoxylin, eosin and safran (HES) to evaluate inflammatory infiltrates (arrows). Immunohistochemistry was conducted using a monoclonal antibody for CVB3/28 viral protein 1 (VP1, orange-brown staining, arrows) of mouse pancreas at 7 DPI (original magnification: ×200). The results shown are representative of 3 mice for each group. (**E**) FL, 8–36 nt and 37–50 nt CVB-TD respective viral loads in mouse pancreases were inoculated with homogenized pancreas from 3, 14 and 28 DPI, or CVB3/28-infected and assessed using a RACE-PCR method associated with micro-electrophoresis from 1 to 28 DPI (*n* = 4 to 8). Data represent mean +/− SEM. Two-way ANOVA test: ** = *p* < 0.01; *** = *p* < 0.001. CVB-FL/TD: full-length or 5′ terminally deleted coxsackievirus. UI: uninfected. DPI: days post-infection.

**Figure 3 viruses-14-02781-f003:**
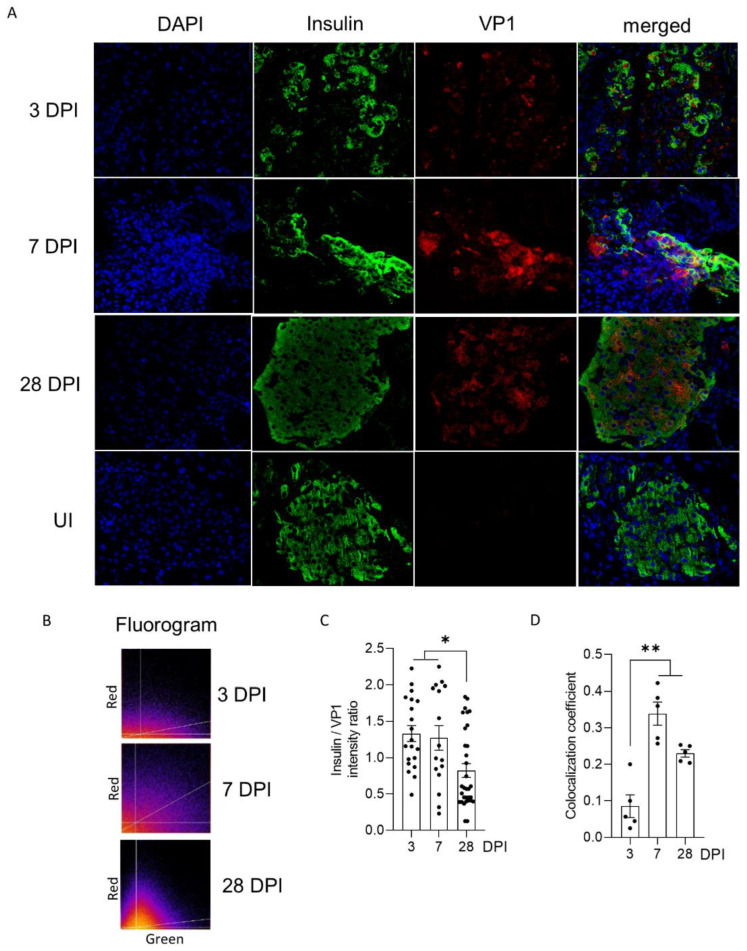
Impact of CVB3/28 infection on the insulin content of pancreatic islet cells of DBA/2J mice. (**A**) An immunofluorescence assay was performed using monoclonal antibody for CVB3/28 VP1 and insulin in thin sections of the pancreas, from 3 to 28 days post-infection (DPI), and from uninfected mice (original magnification: ×200). (**B**) On the fluorograms, green intensities are plotted on the x-axis, and red intensities are plotted on the *y*-axis. The white lines in both plots represent threshold levels applied to both channels for subsequent analysis of the volume. The third line represents a correlation curve using colocalization analyses with JACop. All pixels above and on the right are possible colocalized points. (**C**) Insulin/VP1 intensity ratio was measured for cells positive for insulin from 3, 7 and 28 DPI (*n*= 20, 16 and 32 regions of interest (ROIs) respectively). (**D**) Colocalization coefficient (Pearson) was calculated using QuPath from 3 to 28 DPI in 1 to 3 islets per slice (*n* = 5, each dot represents an islet), from 3 to 28 DPI. The results shown are representative of 3 mice at each indicated time point. ROIs (were chosen within the positive cells for insulin, and colocalization was monitored with the JACoP plugin of ImageJ (manual thresholding). Data represent mean +/− SEM. Mann-Whitney Test. *: *p* < 0.05; **: *p* < 0.01. VP1: viral protein 1. UI: uninfected. DPI: days post-infection.

**Figure 4 viruses-14-02781-f004:**
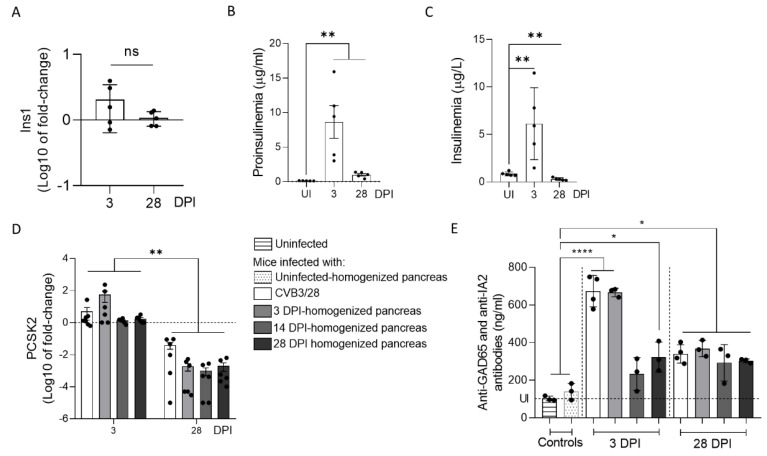
Impact of CVB3/28 and CVB-TD RNA populations’ infections on insulin maturation in pancreases of immunocompetent mouse. (**A**) Insulin (Ins1) mRNA levels’ fold-changes were measured by RT-qPCR in the pancreases of CVB3/28-infected mice at 3 and 28 days post-infection (DPI) (*n* = 5). (**B**) Using specific murine proinsulin ELISA kit, proinsulin levels in blood were measured in uninfected mice and CVB3/28-infected mice at 3 and 28 DPI (*n* = 5). (**C**) Blood insulin was measured using a specific ELISA kit in uninfected and CVB3/28-infected mice at 3 and 28 DPI (*n* = 5). (**D**) PC2 (PCSK2) mRNA levels’ fold-changes were quantified by RT-qPCR in pancreases of CVB3/28-infected mice and of mice inoculated with homogenized pancreases from 3, 14 and 28 DPI (*n* = 5). (**E**) Anti-GAD65 and anti-IA2 antibodies levels in the blood of uninfected mice, CVB3/28-infected mice and of mice inoculated with homogenized pancreases from uninfected, 3, 14 and 28 DPI mice, at 3 and 28 DPI (*n* = 3 to 4) (uninfected mice were sacrificed at 3 or 14 DPI). Data represent mean +/− SEM for three independent experiments. Mann–Whitney U test; *: *p* < 0.05; **: *p* < 0.01; ****: *p* < 0.0001. ns: non-significant. DPI: days post infection. Ins1: preproinsulin gene. PCSK2: Proprotein Convertase Subtilisin/Kexin Type 2. GAD65: glutamic acid decarboxylase 65. IA-2: islet antigen 2.

**Figure 5 viruses-14-02781-f005:**
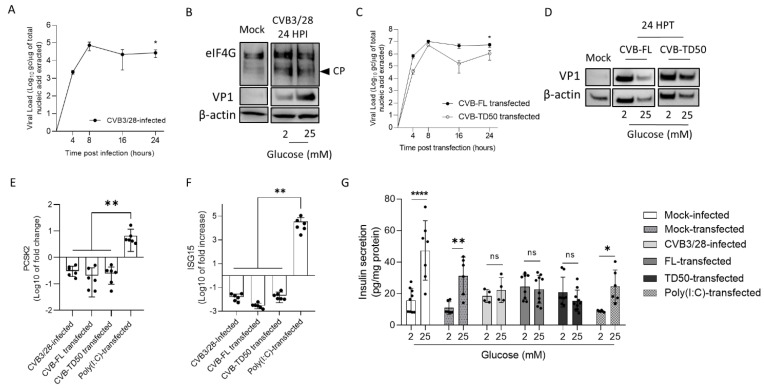
Infection by CVB3/28 and transfection by CVB-TD50 RNA forms decrease PCSK2 mRNA expression and abolish the insulin response to glucose stimulation in beta cells of rodents. (**A**) Infectious CVB3/28 virus loads in the supernatant of INS-1 cells at indicated time points post-infection. (**B**) Western blot analyses showed the cleavage of eIFG4, a eukaryote translation factor, viral protein 1 (VP1) and β-actin expression in mock-infected and CVB3/28-infected pancreases at 24 h post-infection (24 HPI) in INS-1 cells. (**C**) Kinetics of viral RNA replication activity after CVB-FL and CVB-TD50 RNA forms (1 μg) transfection of INS-1 cells measured by RT-qPCR. (**D**) Western blot analysis of CVB3/28 viral protein 1 (VP1) and β-actin in mock-infected and CVB-FL/TD50-transfected INS-1 cells at 24 h post-transfection (24 HPT). (**E**) PC2 (PCSK2) mRNA level fold-changes (RT-qPCR) in INS-1 cells infected with CVB3/28, or transfected with CVB-FL/TD50 or Poly (I:C). (**F**) ISG15 mRNA level fold-changes (RT-qPCR) in INS-1 cells infected with CVB3/28, or transfected with CVB-FL/TD50 or Poly (I:C). (**G**) Glucose stimulated the insulin secretion assay of INS-1 cells following infection of CVB3/28, transfection of CVB RNA forms or Poly (I:C), or mock-infected/transfected cells (triplicate). Data are expressed as pg of insulin in supernatant per mg of total protein content. ANOVA test (panel (**G**)) or Mann–Whitney U test (panels (**A**,**C**,**E**,**F**)); *: *p* < 0.05; **: *p* < 0.01; ****: *p* < 0.0001. Not specified or ns: non-significant. Data represent mean +/− SD of three independent experiments. CVB-FL/TD: full-length or 5′ terminally deleted coxsackievirus. HPI: hours post-infection. HPT: hours post-transfection. CP: cleavage product. VP1: viral capsid protein 1. eIF4G: Eukaryotic translation initiation factor 4G.

**Table 1 viruses-14-02781-t001:** Viral properties of CVB-RNA populations in the pancreas.

		Pancreas		
	Days Post Infection	3	14	28
	(+)/(−) vRNA Ratio	44.9	9.2	2.0
	ss/ds vRNA Ratio	48.2	8.4	2.9
Viral cell-culture challenge (HeLa 229 cells, 72-h)	Entry Capacity (%)	73	13	31
	Proteinase K-treated Entry Capacity (%)	0	0	0
	Fold of normalized genome copy increase	207	4.5	9.3
	PFU (PFU/mL)	9 × 10^6^	ND	ND

N = 3; (+)/(−) or ss/ds vRNA ratio: positive/negative or single/double-strand viral RNA ratio. PFU: plaque-forming unit assay; ND: non-detectable.

## Data Availability

The data presented in this study are available on request from the corresponding author.

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
