# Peer review of "Replication Activities of Major 5′ Terminally Deleted Group-B Coxsackievirus RNA Forms Decrease PCSK2 mRNA Expression Impairing Insulin Maturation in Pancreatic Beta Cells"

_viruses, 2022, doi:10.3390/v14122781_

Round 1

Reviewer 1 Report

Callon et al aimed to investigate the importance of terminally deleted Coxsackieviruses (CVB-TD) in causing pancreas infection and beta cell dysfunction. They confirm their and others recently published studies demonstrating that CVB-TD viruses emerge in the murine pancreas during infection with CVBs (here CVB3/28). Their new data indicate that TD viruses persist over time while infectious "full" viruses are cleared. The authors also investigated the effect of infection by CVB-TD viruses in vivo and in vitro by injecting mice with pancreas material from previously infected mice or transfecting a beta cell line with TD viruses, respectively. Based on their results, they suggest that infection causes a decrease in PC2 mRNA expression and negatively regulate glucose induced insulin secretion. The decreased expression of PC2 could also have a consequence that a higher level of unprocessed insulin (proinsulin) is secreted by cells harbouring TD viruses. The study is of significant interest. There are however a number of aspects including technical aspects which need to be resolved before the manuscript can be accepted. 

1. Please, provide more information on the CVB-TD viruses. Is the terminally deleted viral RNA genome encapsulated in capsids? If not, how are they assumed to infect cells, and in particular the cells of the pancreas following injection of pancreas lysates? Table 1 lists "entry capacity", what does this mean? Please, clarify. 

2. The authors prepared pancreas extracts from animals previously infected by CVB3/28 in order to assess the effect of TD viruses on the pancreas. These extracts were then injected in uninfected animals. Viral load, histology etc were measured at different time points after this. There are a few problems with the approach and questions regarding these experiments:

Firstly, there is no control group in this experiment (animals injected with extracts from uninfected animals). This is a flaw in the design of the study. 

Secondly, the authors perform a histological analysis of pancreases collected from the pancreas extract-injected animals. H&E and VP1 staining were performed. The primary VP1 antibody used is of murine origin, which leads to significant background staining when a secondary anti-mouse antibody is applied to visualise the staining. Please, clarify if any means to eliminate this type of background was used (e.g. by using a mouse on mouse blocking agent or if the primary antibody was directly conjugated). If nothing was done to reduce the background, there is a great risk that unspecific staining confounds the results. Hence, the staining should be repeated using one of these methods.

Thirdly, the morphology of the VP1 stained "UI", "- 14 DPI" or "- 28 DPI" samples is very poor and seems at least in part be due to a lack (or faint) hematoxylin staining. This staining should be improved. 

3. Figure 3 - Also this analysis lacks a proper control (pancreas from mock infected animals). Similar to what was mentioned under #2, whether any measure to eliminate background staining by the secondary anti-mouse antibody should be mentioned alternatively, the studies repeated with such measure. 

4. The authors present dramatically increased anti-GAD65/IA2 antibodies in CVB3/28 infected animals and animals who were injected with pancreas homogenates. This is a very interesting finding. Antibody responses are normally fairly slow and it is rare to see such high antibody levels already at day 3 post infection or other exposure (note, not even virus neutralising antibodies are present this early after infection), what are the authors thoughts around this? Please, discuss in the text. 

5. Font text in the figures could be enlarged for increased readability.

Reviewer 2 Report

In this manuscript Callon et al have looked at the generation of terminally deleted Coxsackievirus B3 (CVB) variants in the pancreas of mice infected with full-length CVB3 in an immunocompetent mouse model. The mice were susceptible to infection and terminally deleted CVB variants were detected in the pancreas at acute (day 3) and persistent (days 14 and 28) times after infection. Moreover terminally deleted variants isolated from the pancreas were able to infect other mice causing further acute and chronic infections highlighted by the presence of viral RNA and the expression of virus proteins. The pancreas of these infected animals also had lower pro-hormone convertase 2 mRNA expression and decreased plasma insulin concentrations. Synthetic CVB-TD viruses were able to decrease PCK2 expression in cultured rodent beta-cells where they also impaired insulin secretion. Taken together, this study suggests a mechanism through which CVB infection may lead to the development of type 1 diabetes. This paper sets out to ask an important question regarding how persistent CVB infections may lead to the development of type 1 diabetes and for the most part the results described provide an indication of what may happen. However further studies/data is required for some parts to strengthen the conclusions made.

The paper is generally clear and well written although improvements in the language used is required.

Comments:

Methods section:

-       Line 96: Should be Dr Steve Tracy not Dr Steve Stracy

-       Section 2.1: Have the cell lines been tested for their mycoplasma status and have the cell lines been tested to confirm they are what they are supposed to be?

-       Sections 2.3 and 2.4: More information is required regarding the breeding of the mice, housing conditions, monitoring etc.

-       Section 2.5: Two different VP1 antibody dilutions are given in the methods section, which is correct? The VP1 antibody clone used is raised in mouse, how have the authors ensured that there is no non-specific binding with the secondary antibodies used and that the staining seen is true VP1 staining? Which dilutions of secondary antibody are used?

-       Section 2.6: Line 155 – please include the section (e.g. section 2.2) rather than “as described above”. 

-       Section 2.7: please expand on the infection protocol – how long were the INS-1 cells infected for, were the cells washed after infection, was the infection media replaced? Line 163 – where are the transfection experiments previously described?

-       Section 2.8: Which method was used for measuring insulin secretion?

-       Section 2.9: Which antibody concentrations were used?

-       Section 2.12: Line 220-221 – please rephrase this sentence as it doesn’t make sense.

-       Section 2.15: Line 246 – please give the section where the methods were described rather than as described above. 

Results section:

-       Section 3.1: Have you sequenced the CVB3/28 stock to check that there are no terminally deleted variants present?

-       Section 3.2: Lines 298-301 – please expand on the encapsidation experiments, the theory behind the proteinase K treatment isn’t so clear.

-       Section 3.2: Lines 301-303 – how is VP1 synthesis being shown in Table 1 and/or figure 1? 

-       Figure 2A: please change the colour palette so that it isn’t red/green for those that are colour blind.

-       Figure 2B, C: are there any statistics for this? Do the symbols represent individual mice?

-       Figure 2D: 

o   The quality of the IHC images needs to be improved as some of the images aren’t clear (either on the computer or printed out). 

o   No scale bars are shown and are all of the images the same magnification?

o   Were the images in the VP1 panel counterstained with haematoxylin? It looks like they were in the CVB3/28 and 3DPI panels but not in the other panels. 

o   It would be interesting to see quantification of the VP1 positivity in the sections (e.g. area that is positive, number of islets positive for VP1 and the number of VP1 positive cells per islet. Same for the pancreatitis scoring.

-       Figure 2E: Are you able to speculate as to why the viral loads of the CVB-TD variants are lower in the day 14 homogenized pancreas than the day 3 homogenized pancreas given that in figure A they are present to a higher degree? 

-       Section 3.3: Lines 326-329: these conclusions are a bit too strong in my opinion. There appears to be VP1 positivity in the islets but no evidence that it is specifically in the beta-cells from the IHC. 

-       Figure 3A: 

o   Please include a control non-infected mouse pancreas showing “normal” insulin staining and the intensity.

o   Why does the intensity of the DAPI staining vary so much between the different time points. It is much more intense in the 7 DPI row compared to the 3DPI and 28DPI rows which may account for differences in the insulin/VP1 intensity.

o   What are the lines in the fluorogram showing? Please describe in the figure legend.

o   How were the islets selected for the analysis described? What proportion of islets were positive for VP1? Some of the cells that stained positive for VP1 appear to be non-beta-cells, have you performed staining to see which cells these are?

-       Section 3.4: Is it possible to stain for pro-insulin and insulin separately? It would be interesting to see if there are differences in staining pattern in the beta-cells after infection. It would also be interesting to see IHC/IF staining of PCSK2 in the pancreas after infection to support the mRNA data shown in Figure 4 D. 

-       Section 3.4: Lines 379-382 – again this statement seems too strong, it may be that the CVB3/28 and CVB-TD viruses cause the impairment of proinsulin to insulin maturation but it is still speculative given the results. Also, there is no proof that the autoantibodies are themselves causing the decrease in insulin secretion in this paper.

-       Section 3.4: Did you look at the blood glucose levels of the mice throughout the time period studies? It would be interesting to see if there were signs of hyperglycaemia in the animals infected with the pancreas homogenates from the different time points after infection. 

-       Section 3.5: Lines 409-411 – the sentence needs to be re-written as currently unclear.

-       Section 3.5: Lines 424-426 – the sentence needs to be re-rewritten as currently unclear.

-       Figure 5B: why does there appear to be some cleavage product of eIF4G in the mock lane?

-       Figure 5D: why is there a faint VP1 band in the mock?

-       Figure 5E,F: It would be interesting to see Western blot data for PCSK2 and ISG15 if possible to confirm the mRNA data. 

-       Figure 5G: It appears that transfection of the INS-1 cells decreases insulin secretion itself. Please include a control in the transfection experiments to show whether this occurs as it makes it harder to come to the conclusion that the insulin secretion response is functional in transfected cells. Also there is a question mark above the poly(I:C)-transfected columns. 

Discussion section:

-       Lines 468-471: Is the evidence of pancreatitis clear enough to come to this conclusion?

-       Line 474: please expand on this sentence and explain how you saw this.

-       Lines 489-492: given the speculation surrounding the involvement of TBP, it would be interesting to see Western blot data for this protein too in a similar manner to eIF4G in figure 5.

-       Lines 543-549: A CVB vaccine is currently undergoing clinical trial in Finland (Provention Bio) which focuses on the involvement of CVBs in diabetes.

Other comments:

-       It would be interesting to see the in vitro data applied to a human model e.g. a human beta-cell line (EndoC-BH1) or primary human islets where possible.

Figure Legends:

Please provide a clearer description of the figures in the legend e.g. expand on all abbreviations present in the figures in the legends.

Round 2

Reviewer 2 Report

Please find attached comments (in red) to the authors response.

Author Response

Please see the attachment (responses in blue)
